# Pathological pallidal beta activity in Parkinson's disease is sustained during sleep and associated with sleep disturbance

Zixiao Yin [1], Ruoyu Ma [1], Qi An[1], Yichen Xu[1], Yifei Gan[1], Guanyu Zhu[1], Yin Jiang[2], Ning Zhang[3], Anchao Yang[1], Fangang Meng[2], Andrea A. Kühn [4,5], Hagai Bergman [6,7,8], Wolf-Julian Neumann [4,10] ✉ & Jianguo Zhang [1,2,9,10] ✉

Parkinson's disease (PD) is associated with excessive beta activity in the basal ganglia. Brain sensing implants aim to leverage this biomarker for demand-dependent adaptive stimulation. Sleep disturbance is among the most common non-motor symptoms in PD, but its relationship with beta activity is unknown. To investigate the clinical potential of beta activity as a biomarker for sleep quality in PD, we recorded pallidal local field potentials during polysomnography in PD patients off dopaminergic medication and compared the results to dystonia patients. PD patients exhibited sustained and elevated beta activity across wakefulness, rapid eye movement (REM), and non-REM sleep, which was correlated with sleep disturbance. Simulation of adaptive stimulation revealed that sleep-related beta activity changes remain unaccounted for by current algorithms, with potential negative outcomes in sleep quality and overall quality of life for patients.

Sleep disturbances significantly impair the quality of life in people living with Parkinson's disease (PD)[1]. Insomnia and rapid eye movement (REM) sleep behavior disorder (RBD) are common manifestations of PD-related sleep disturbance[2,3] that can be correlated and sometimes even predict other non-motor symptoms, such as cognitive decline[4,5]. Current therapies including medication and continuous deep brain stimulation (DBS) are not sufficiently adjustable to specifically target sleep disturbance in PD[6]. This represents an unmet need in the development of individualized therapeutic approaches. Adaptive DBS is a novel treatment strategy that promises unprecedented temporal and spatial precision for therapeutic adjustment[7]. This opens new horizons for circadian and sleep-related therapeutic adaptation[8,9].

First technological solutions to sleep-aware adaptive DBS and chronotherapeutic treatment strategies have now emerged, with the potential to revolutionize the targeted treatment of sleep disturbance[8]. However, the first ongoing clinical trials are primarily informed by data obtained from awake patients. Specifically, excessive beta activity, a hallmark of the hypodopaminergic parkinsonian state, is being used as a biomarker for adaptive DBS[10] and was found to be reduced during sleep[11–15]. While modulation of beta activity was reported during sleep cycles, the pathophysiological impact of this activity pattern on sleep quality in human patients remains unknown. Recently, a potential relationship between beta activity and sleep disturbance has first been reported in a non-human primate model of Parkinson's

[1]Department of Neurosurgery, Beijing Tiantan Hospital, Capital Medical University, Beijing, China. [2]Department of Functional Neurosurgery, Beijing Neurosurgical Institute, Capital Medical University, Beijing, China. [3]Department of Neuropsychiatry, Behavioral Neurology and Sleep Center, Beijing Tiantan Hospital, Capital Medical University, Beijing, China. [4]Department of Neurology, Movement Disorders and Neuromodulation Unit, Charité – Universitätsmedizin Berlin, Charitéplatz 1, 10117 Berlin, Germany. [5]Exzellenzcluster – NeuroCure, Charité – Universitätsmedizin Berlin, Berlin, Germany. [6]The Edmond and Lily Safra Center for Brain Sciences, The Hebrew University, Jerusalem, Israel. [7]Department of Medical Neurobiology (Physiology), Institute of Medical Research – Israel Canada (IMRIC), Faculty of Medicine, The Hebrew University, Jerusalem, Israel. [8]Department of Neurosurgery, Hadassah Medical Center, Jerusalem, Israel. [9]Beijing Key Laboratory of Neurostimulation, Beijing, China. [10]These authors contributed equally: Wolf-Julian Neumann, Jianguo Zhang. ✉e-mail: julian.neumann@charite.de; zjguo73@126.com

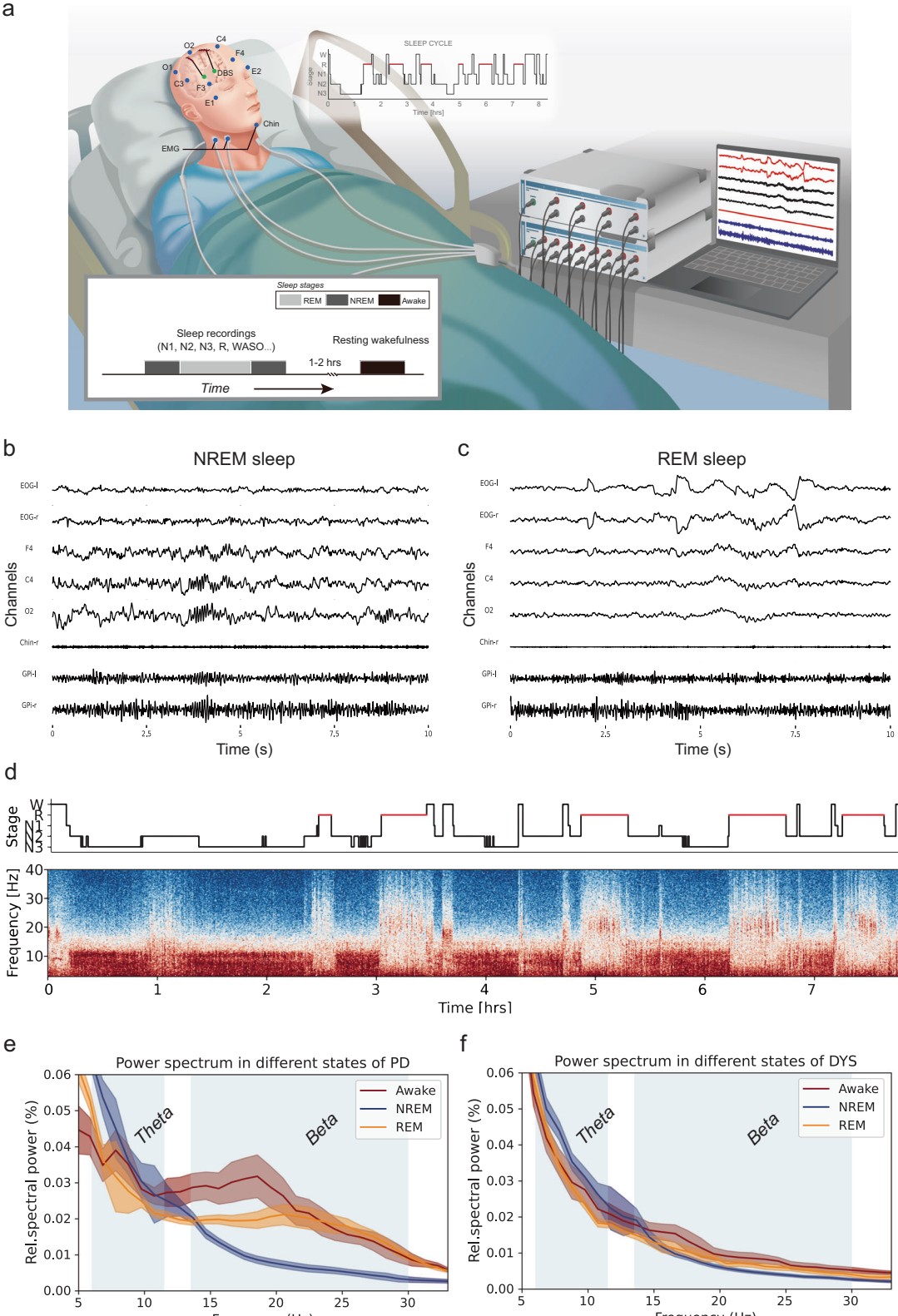

disease[16], urging further investigation in the human domain. With the present study, we aim to address this important knowledge gap by comparing pathological beta activity across sleep cycles in the internal pallidum from patients with Parkinson's disease with a control group of subjects suffering from dystonia, a different neurological disorder treated with internal pallidum DBS.

## Results

Twelve subjects with Parkinson's disease and twenty subjects with dystonia undergoing DBS electrode implantation in the ventro-posterior-lateral (motor) domain of the internal pallidum (GPi) were recruited for electrophysiological recordings during sleep. Standardized polysomnography was combined with invasive local field

**Fig. 1 | Recordings of pallidal local field potentials in parallel to polysomnography in subjects with Parkinson's disease and dystonia. a** Schematic representation of the sleep recording. Pallidal local field potentials are recorded in parallel to the polysomnography consisting of the electroencephalogram (EEG), electrooculogram (EOG), and electromyogram (EMG). In the morning after sleep recording, a 5-min recording of resting wakefulness is further obtained. **b** shows 10 s of characteristic N2 sleep from subject PD-8, epoch 26. Characteristic sleep spindles are seen in polysomnography. Note that for visualization, the amplitude of the pallidal channels is amplified. **c** shows 10 s of characteristic REM sleep from subject PD-8, epoch 16. Prominent rapid eye movement, low-voltage waves, and muscle atonia are seen in EOG, EEG, and EMG, respectively. **d** shows a representative spectrogram of a whole-night recording from subject PD-8 with the hypnogram on top. **e** shows the average power spectra across awake, NREM, and REM sleep epochs in all patients with Parkinson's disease. **f** shows the average power spectra across awake, NREM, and REM sleep epochs in all patients with dystonia. Shaded areas represent SEM. Theta and beta frequency band ranges are highlighted in blue.

potential (LFP) recordings from the DBS electrodes in the GPi (Fig. 1a). After visual and algorithmic sleep staging, data from 40 nights across 32 patients were analyzed (for clinical information see Table 1). A comparison of sleep parameters indicated that patients with PD had significantly shorter total sleep time, longer sleep latency, less REM sleep, and more sleep segmentations than patients with dystonia (Table 2). Representative non-REM (NREM) and REM sleep epochs as well as a whole-night hypnogram and the corresponding spectrogram (from subject PD-8) are shown in Fig. 1b–d. Average power spectra corroborate previous reports indicating reduced beta activity during non-REM[11–14], when compared to awake and REM stages (Fig. 1e, f). However, these studies did not compare beta activity during sleep to a control group without Parkinson's disease.

### Pallidal beta activity in PD significantly exceeds controls in NREM and REM sleep

To study the pathophysiological significance of beta activity during NREM and REM stages we compared pallidal beta activities to the control group of patients with dystonia (Fig. 2). First, we reproduced previous reports in awake patients demonstrating higher beta activity in PD ($P = 0.001$, Mann–Whitney $U$ test) and higher theta activity in dystonia patients ($P < 0.001$, Mann–Whitney $U$ test). Importantly, excessive beta activity was not restricted to the awake stage but sustained during NREM ($P = 0.021$, Mann–Whitney $U$ test) and REM ($P = 0.001$, Mann–Whitney $U$ test) sleep epochs. Substage analysis of NREM sleep indicated that beta power was higher in patients with PD in N1 ($P = 0.006$) and N2 ($P = 0.013$), but not in N3 sleep ($P = 0.093$, Mann–Whitney $U$ test) than in patients with dystonia (Supplementary Fig. 1). To control for potential confounds due to changes in the aperiodic component of the spectra, we performed a supplementary comparison of the periodic activity after spectral parameterization[17], which confirmed higher beta activity in PD across all stages (Supplementary Fig. 2). Further, beta burst analysis indicated that the excessive beta activity during NREM and REM sleep in PD could be attributed to prolonged burst durations (Supplementary Fig. 3). The significant differences were specific to the pallidal recordings, with activity spatially peaking within 1.50–2.06 mm proximity to a previously reported optimal stimulation target location[18]. No difference in beta activity was present in cortical electroencephalography (Supplementary Fig. 4).

### Excessive pallidal beta activity during NREM sleep is associated with lower sleep quality

Following a previous non-human primate study[16] we hypothesized that pathological beta activity during sleep may be associated with PD-related sleep disturbance. To investigate this, we correlated pallidal beta power during NREM and REM episodes with sleep quality ratings (Fig. 3). Pallidal beta power during NREM sleep was robustly correlated with the Pittsburgh sleep quality index (PSQI) in PD (Spearman $rho = 0.63$, $P = 0.028$) but not dystonia patients (Spearman $rho = 0.19$, $P = 0.434$, Fig. 3a). This correlation was more robust for high beta (20–30 Hz) than low beta (13-20 Hz) power, and for NREM2 than NREM1/3 stages of sleep (Fig. 3b). A significant correlation was identified between beta power during NREM sleep and the RBD-Screening Questionnaire (RBDSQ) score, but this finding may have limited specificity as no correlation was found when analyzing the proportion of REM sleep time without atonia as a proxy of RBD severity (Supplementary Fig. 5). Finally, PD motor sign severity as assessed with the motor part of the Unified Parkinson's disease rating scale (UPDRS-III) was correlated with pallidal beta activity in awake (Spearman $rho = 0.78$, $P = 0.003$) but not sleep epochs (all $P > 0.05$). For a summary of the clinical correlation results see Fig. 3c.

### Common adaptive DBS algorithms will restrain therapeutic delivery during sleep while pathological activity may be sustained

Given the abovementioned results indicating higher pallidal beta activity in PD than in dystonia across REM and NREM epochs, but lower beta activity when compared to awake stages, we hypothesized that common adaptive DBS algorithms may not sufficiently respond to pathological beta activity during sleep. This may be problematic, especially when beta activity is associated with sleep disturbance. To address this, we simulated a common threshold-based (i.e., 50th percentile[19]) adaptive DBS algorithm calibrated on the awake data (Fig. 4). Our results suggest that adaptive DBS would reduce therapeutic deliveries during NREM (stimulation-on time: 4.98%) and REM (stimulation-on time: 30.00%) sleep stages, which was independent of a potential effect of sleep duration.

## Discussion

Three main conclusions can be drawn from our study. First, we show evidence that pallidal beta activity is elevated across NREM and REM sleep in PD when compared to recordings from patients with dystonia. This result could be attributed to longer burst durations and was specific to a localized peak in proximity to the optimal stimulation target in the internal pallidum and absent in cortical EEG recordings. Second, our findings suggest that specifically elevated beta activity during NREM sleep, especially the N2 sleep stage, can be associated with sleep disturbance and lower sleep quality in PD but not dystonia patients. Finally, we demonstrate that common adaptive DBS algorithms calibrated on awake data will not modulate pathological beta activity during NREM and REM sleep stages. This is particularly important as it may be addressed through refined brain signal decoding algorithms in sleep-aware adaptive DBS paradigms. Our findings could pave the way for an individualized treatment of sleep disturbance in PD.

Our findings corroborate an impactful report linking beta activity with sleep disturbance in the 1-methyl-4-phenyl-1,2,3,6-tetrahydropyridine (MPTP) non-human primate model of PD[16]. The study demonstrated that MPTP intoxication induced elevated beta power and spiking during NREM sleep in the subthalamic nucleus and external and internal pallidum. This was associated with delayed sleep onset, increased sleep fragmentation, and increased wakefulness in non-human primates when compared to the healthy state. In human studies, such direct comparisons to healthy states are not possible, because invasive recordings are only available in patients undergoing neurosurgical interventions for brain disorders. To relate neural activity to disease-specific aspects, two strategies have emerged: (a) relative differences in activity from the same anatomical structure can be compared to control groups with other brain disorders[20–22] and (b) within cohort correlations may indicate associations of brain activity

**Table 1 | Demographics and clinical characteristics of the included patients**

| Patient | Nights (n) | DD (y) | PSQI | RBDSQ | Motor score[a] | Medication |
|---|---|---|---|---|---|---|
| *Parkinson's disease* | | | | | | |
| PD-1 | 1 | 20 | 16 | 11 | 46/28 | Madopar, Amantadine, Piribedil |
| PD-2 | 1 | 7 | 15 | 10 | 26/11 | Madopar, Rasagiline |
| PD-3 | 1 | 6 | 9 | 2 | 53/23 | Madopar, Sinemet, Pramipexole |
| PD-4 | 1 | 6 | 13 | 5 | 33/21 | Madopar, Piribedil, Entacapone, Pramipexole |
| PD-5 | 1 | 5 | 6 | 1 | 29/15 | Madopar |
| PD-6 | 1 | 8 | 9 | 1 | 45/31 | Madopar, Piribedil |
| PD-7 | 2 | 10 | 13 | 7 | 69/31 | Madopar, Sinemet |
| PD-8 | 2 | 6 | 6 | 1 | 71/32 | Sinemet, Piribedil, Entacapone |
| PD-9 | 2 | 12 | 15 | 2 | 54/18 | Madopar, Sinemet, Amantadine |
| PD-10 | 1 | 9 | 12 | 12 | 67/32 | Madopar, Trihexyphenidy, Piribedil |
| PD-11 | 1 | 7 | 5 | 1 | 45/23 | Madopar |
| PD-12 | 2 | 8 | 17 | 13 | 66/37 | Madopar, Entacapone, Pramipexole |
| Median (IQR) | | 7.5 (4.0) | 15.0 (7.0) | 3.5 (9.5) | 49.5 (30.8)/25.5 (13) | |
| *Dystonia* | | | | | | |
| Dyst-1 | 2 | 2 | 5 | – | C/20 | Benzhexol, Baclofen, Clonazepam |
| Dyst-2 | 2 | 3 | 10 | – | M/8 | Clonazepam, Tiapride hydrochloride, Mecobalamin |
| Dyst-3 | 1 | 5 | 9 | – | M/7 | Botulin, Carbamazepine |
| Dyst-4 | 2 | 5 | 9 | – | M/23 | Clonazepam |
| Dyst-5 | 1 | 4 | 6 | – | M/6 | Baclofen |
| Dyst-6 | 1 | 4 | 1 | – | C/28 | Amantadine, Benzhexol |
| Dyst-7 | 1 | 3 | 19 | – | M/12 | Tiapride hydrochloride |
| Dyst-8 | 1 | 5 | 14 | – | M/13 | Tiapride hydrochloride |
| Dyst-9 | 1 | 7 | 5 | – | M/16 | Botulin |
| Dyst-10 | 2 | 3 | 3 | – | C/32 | NA |
| Dyst-11 | 1 | 15 | 11 | – | M/18 | Botulin |
| Dyst-12 | 1 | 15 | 1 | – | M/7 | NA |
| Dyst-13 | 1 | 4 | 5 | – | C/22 | Baclofen, Clonazepam |
| Dyst-14 | 1 | 3 | 3 | – | M/9 | Clonazepam |
| Dyst-15 | 1 | 5 | 11 | – | M/15 | Benzhexol, Haloperidol |
| Dyst-16 | 1 | 6 | 7 | – | C/38 | NA |
| Dyst-17 | 1 | 7 | 6 | – | M/13 | Botulin, Tiapride hydrochloride |
| Dyst-18 | 1 | 3 | 6 | – | M/16 | Botulin |
| Dyst-19 | 1 | 6 | 10 | – | C/24 | Clonazepam |
| Dyst-20 | 1 | 2 | 6 | – | C/17 | Baclofen, Haloperidol |
| Median (IQR) | | 4.5 (3.0) | 6.0 (5.0) | – | | |

*DD* duration of disease, *PSQI* Pittsburgh sleep quality index, *RBDSQ* REM sleep behavior disorder-screening questionnaire, *Dyst* dystonia, *PD* Parkinson's disease, *C* cervical dystonia, *M* Meige syndrome (oromandibular dystonia), *IQR* interquartile range, *NA* not applicable.
[a]Preoperative motor score was the Toronto Western Spasmodic Torticollis Rating for cervical dystonia, Burke–Fahn–Marsden Dystonia Rating Scale (movement) for oromandibular dystonia, and MDS-Unified Parkinson's Disease Rating Scale-III off/on medication for Parkinson's disease.

patterns with clinical signs of the disease[23]. In the awake state, both of these strategies have demonstrated a clear association between beta activity and PD motor signs, where basal ganglia beta activity is elevated when compared to patients with dystonia[20,21] and robustly associated with motor sign severity, most recently reproduced in a cohort of 106 patients[24]. We have now extended both approaches to the sleep state, for which previous studies[11–15] have demonstrated that beta is influenced by sleep-stage transitions, promising potential utility for sleep-stage decoding[25], but the relative pathological significance remained unaddressed. Our study extends these insights by providing direct evidence that compared to dystonia subjects, pallidal beta activity in PD is continuously higher across sleep stages and correlated with sleep disturbance.

As previously reported, we found a drop in beta activity during NREM stages, when compared to REM and wakefulness, resulting in a relative overlap of power spectra across the PD and the dystonia groups. However, despite the amplitude decrease, our results suggest that beta power remains excessively high during NREM sleep when comparing PD with dystonia subjects. In addition, we found evidence that this pathophysiological pattern during the NREM phase may be associated with impaired sleep quality, as beta power during NREM sleep showed robust correlations with the PSQI, a validated assessment of sleep disturbance. This correlation was most robust in the NREM2 stage of sleep, potentially because N2 sleep occupied the longest duration of sleep time (over 60% in our patient cohort) and that physiologically important sleep oscillations such as spindles and K-complexes are typically most prominent in N2 sleep[26]. When comparing beta power across sleep cycles in the PD cohort, we found a relative decline in beta power from N1 sleep, where it was close to that in wakefulness, to N3 sleep, where it was more similar to that in dystonia. Whether this indicates a potential "normalization" effect of N3 sleep and whether, inversely, a suppression of basal ganglia beta activities could result in longer deep sleep in PD requires further investigation. When assessing the spectral specificity of our results, we

**Table 2 | Sleep parameters of the included patients[a]**

| | Parkinson's disease (n = 12) | Dystonia (n = 20) | P values[b] |
|---|---|---|---|
| TTB (min) | 494.0 (51.2) | 509.0 (59.0) | 0.599 |
| TST (min) | 298.4 (108.2) | 368.2 (75.0) | **0.012** |
| WASO (min) | 222.0 (143.0) | 138.4 (55.0) | 0.129 |
| SL (min) | 50.1 (55.7) | 14.2 (23.7) | **0.049** |
| RSL (min) | 206.4 (181.3) | 108.5 (77.0) | **0.042** |
| N1pct (%) | 8.5 (14.9) | 7.1 (7.5) | 0.800 |
| N2pct (%) | 60.6 (15.4) | 65.8 (17.5) | 0.302 |
| N3pct (%) | 14.9 (10.9) | 8.1 (10.6) | 0.115 |
| Rpct (%) | 13.0 (6.0) | 16.1 (9.8) | **0.037** |
| SE (%) | 59.9 (29.6) | 74.3 (12.9) | 0.064 |
| Sfrag (n)[c] | 15.5 (8.8) | 9.5 (5.2) | **0.015** |
| RSWA (%)[d] | 26.1 (24.9) | 7.8 (14.8) | 0.083 |

*TTB* total time in bed, *TST* total sleep time, *WASO* wake after sleep onset, *SL* sleep latency, *RSL* REM sleep latency, *N1pct* percentage of the NREM stage 1 sleep, *N2pct* percentage of the NREM stage 2 sleep, *N3pct* percentage of the NREM stage 3 sleep, *Rpct* percentage of the REM sleep, *SE* sleep efficiency, *Sfrag* sleep fragmentations, *RSWA* REM sleep without atonia.

[a]Descriptive data are presented as the median (interquartile range).

[b]P Statistics are obtained using two-sided Mann–Whitney *U* test without applying the Bonferroni correction. Significant comparisons with uncorrected *P* value < 0.05 are highlighted in bold.

[c]Sleep fragmentation is quantified as the times that sleep is interrupted by >2 minutes' wakefulness.

[d]RSWA is quantified as the percentage of REM sleep time when EMG activities are higher than two times the 5th percentile of the chin EMG activities during NREM sleep (detailed in the "Methods" section).

noted that high beta (20–30 Hz), rather than low beta (13–20 Hz) power, showed robust correlations with sleep disturbances. In the awake state, especially in the subthalamic nucleus, low beta activity was more robustly modulated by therapeutic interventions and correlated with PD motor signs[27]. Yet, even in awake recordings in the present cohort, high beta ($rho = 0.715$, $P = 0.009$), rather than low beta ($rho = 0.410$, $P = 0.186$) power correlated most strongly with UPDRS-III. We speculate that a) the relative relationship of spectral frequency and clinical phenomena may be subject-specific, and (b) a potential systematic difference may arise from the fact that the presented activity patterns were recorded from the internal pallidum, while most previous work described activity from the subthalamic nucleus. In summary, our study provides evidence that pallidal beta activity may play a pathological role in PD both during sleep and in wakefulness.

One of the most robust prodromal signs of alpha-synuclein neuropathology including Parkinson's disease is REM sleep behavior disorder, a parasomnia associated with dream enactment and loss of REM sleep atonia[28]. In the present cohort, clear dream enactment behavior was not observed in any of the included PD patients, but NREM but not REM-related beta activity was found to be associated with higher scores in the RBD-Screening Questionnaire. Given that this questionnaire is unspecific and does not provide a continuous scaling of RBD severity, we further quantified the loss of REM sleep atonia, as an indirect marker of RBD severity. We observed REM sleep without atonia in a median of 26.1% of all REM sleep epochs in PD patients but found no significant correlation between the relative proportion of these epochs with beta activity measures. Thus, considering the limited sensitivity and specificity of the RBD-Screening Questionnaire in diagnosing REM sleep behavior disorder[29], we speculate that the correlation between NREM beta power and RBDSQ could rather corroborate our abovementioned finding that beta can generally index sleep disturbances, which is further substantiated by the high correlation between PSQI and RBDSQ scores ($rho = 0.821$, $P = 0.001$). A recent, more specific investigation of four PD subjects proposed that pallidal beta oscillations can be synchronized during movements in the REM sleep phase[30]. Future studies with expanded sample size should aim for a direct comparison of patient cohorts with/without REM sleep

behavior disorder, optimally capturing the neural dynamics of the basal ganglia while parasomnia is present. This could shed light on the relevance of pathological beta activity in dream enactment, potentially even in the prodromal stages of Parkinson's disease.

Sleep quality in PD patients can be improved with DBS across basal ganglia targets (including subthalamic nucleus[31], internal pallidum[32], and external pallidum[33]), but the therapeutic mechanism remains elusive[34]. Our results suggest that excessive beta power in NREM sleep could be a therapeutic target to treat sleep disturbance. Moreover, given that DBS, similar to medication can suppress beta activity[35], we may speculate that sleep improvement through both treatments may be associated with beta reduction. Future studies should verify whether high-frequency stimulation applied during NREM and REM sleep has the same effect on beta suppression as that applied during wakefulness. Furthermore, the best stimulation pattern required to suppress beta during sleep remains to be determined. Since we did not identify sleep-related behavior correlates of beta activity during the REM stage here, and given evidence showing that DBS may occasionally induce de novo RBD[36], it requires further investigation on whether stimulation should be switched on or off during REM sleep.

The direct relationship of beta activity with PD motor signs has inspired a new adaptive DBS treatment paradigm, that uses beta activity as a biomarker to adapt stimulation to therapeutic demand[37]. A challenging aspect of this novel therapeutic approach is the definition of a valid control algorithm, that translates neurophysiological recordings into stimulation parameter changes. Notwithstanding this, the first international multicenter trials are now investigating the potential clinical utility of this approach. While specific algorithms can vary, a common denominator across centers and studies is the fact that one or multiple thresholds are required to calibrate the control algorithm[38]. To date, these thresholds are being determined while patients are awake. In our study, we demonstrated that even though beta activity is significantly reduced during NREM, it remains excessive when compared to dystonia. This may indicate that DBS during NREM stages may be necessary for the alleviation of sleep disturbance in PD. Our simulation of a common algorithm, however, revealed that excessive NREM beta activity may remain untreated in the ongoing adaptive DBS trials. Adopting the conventional threshold-based approach, adaptive DBS would have only triggered in around 5% of the NREM and 30% of REM sleep phases when maintaining at least 50% of stimulation time during wakefulness. We believe it is important to raise awareness that this new therapy has very specific sleep stage-dependent effects that should be closely studied in clinical trials. Moreover, our findings call for novel sleep-aware adaptive DBS implementations, that can automatically dissociate REM and NREM stages and adapt thresholds and stimulation delivery accordingly. The feasibility of this approach was recently shown in a sensing-enabled impulse generator using a combination of two machine learning classifiers trained on electrocorticography and LFP recordings[8]. While generally very promising, it is important to note, that pathophysiological phenomena may interact with such machine learning algorithms, as recently shown for beta bursts that can detriment grip-force decoding in PD[39]. In the future, sleep may be one of many decoding targets[25], through which machine learning will extend the clinical utility of adaptive DBS adjusting to the individual challenges of our patients in their everyday lives[40–42]. Such intelligent adaptive DBS systems may not only improve the nocturnal motor symptoms of PD but also address sleep and sleep-related dysfunctions in PD as a whole.

We would like to highlight the following limitations of our study. First, even though the number of included subjects in this study is higher than most previous studies with similar aims[14,30] the sample size is still relatively small. Second, subjects suffering from poor sleep quality had reduced or even completely abolished REM sleep[14,16], which further reduces the amount of data for REM sleep analyses in this

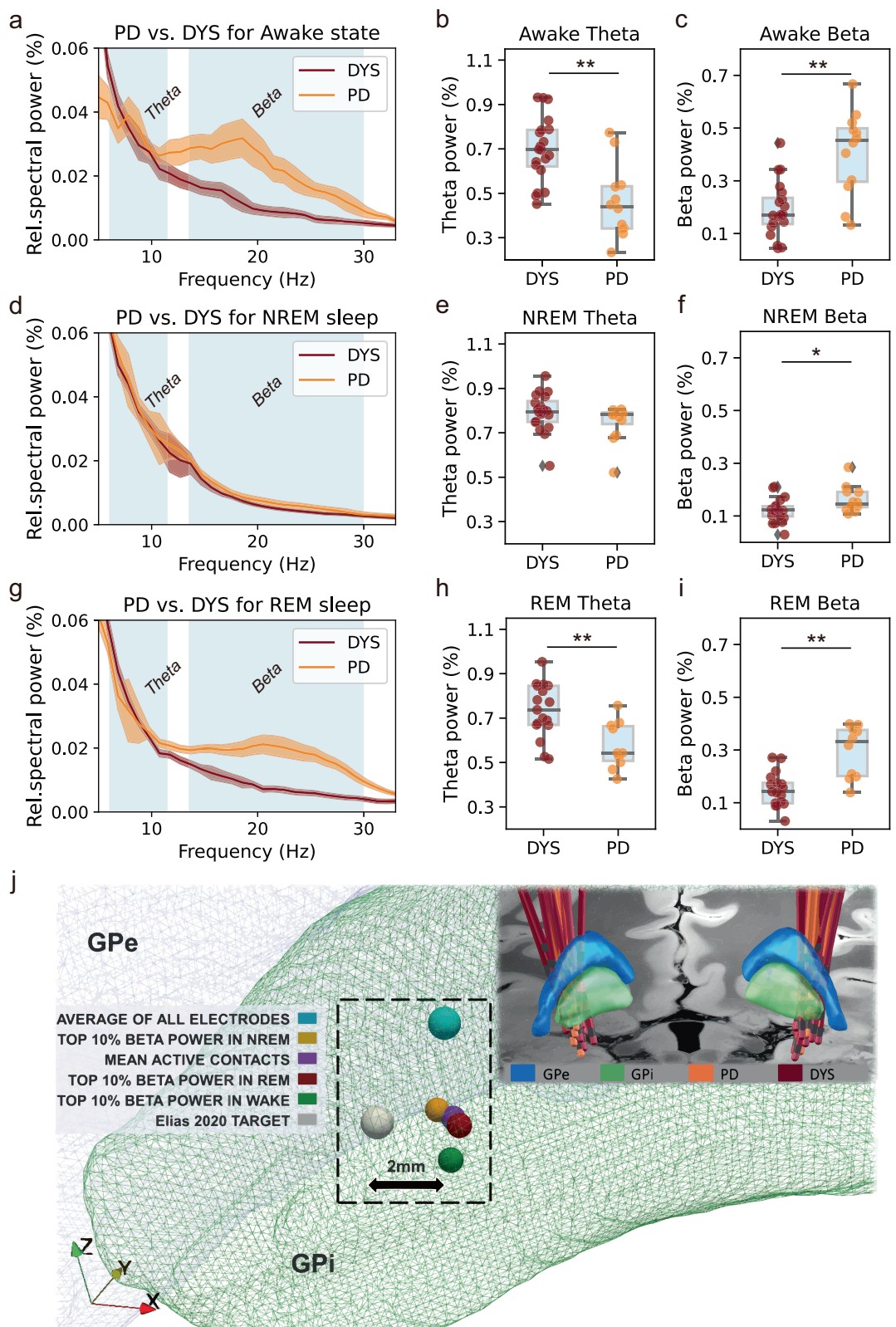

study. Recordings across multiple nights could be a potential solution to this problem. Third, our data are recorded through externalized leads rather than sensing-enable devices, which may in theory enable longer recording periods[15] and provide a more ecologically valid in-house assessment of sleep. However, current commercially available devices, such as the Medtronic Percept, can only capture a single value of beta power per hemisphere every ten minutes. Thus, externalized recordings have unique advantages including higher sampling rate and better synchronization between LFP signals and PSG[43], which is especially important for sleep studies. A fourth limitation of the present study is the focus on excessive beta synchronization in PD. Future investigations may lay more focus on increased pallidal theta power during REM sleep in dystonia to expand our understanding of disease-specific oscillatory abnormalities in the basal ganglia during sleep.

**Fig. 2 | Comparisons of power spectra between Parkinson's disease (PD) and dystonia across different sleep stages and the spatial localization of beta in PD. a–c** Power spectra and comparisons of theta and beta power in awake epochs between PD and dystonia. **\*\****P* for theta power<0.001; \*\**P* for beta power=0.001; *n* for PD subjects = 12; *n* for dystonia subjects = 20; two-sided Mann–Whitney *U* test. **d–f** Power spectra and comparisons of theta and beta power in non-rapid eye movement (NREM) sleep between PD and dystonia. *P* for theta power = 0.083; \* *P* for beta power = 0.021; *n* for PD subjects = 12; *n* for dystonia subjects = 20; two-sided Mann–Whitney *U* test. **g–i** Power spectra and comparisons of theta and beta power in REM sleep between PD and dystonia. \*\**P* for theta power = 0.004; \*\**P* for beta power = 0.001; *n* for PD subjects = 10; *n* for dystonia subjects = 17; two-sided Mann–Whitney *U* test. Shaded areas in all spectrum plots represent SEM. For all box plots, the lower and upper borders of the box represent the 25th and 75th percentiles, respectively. The centerline represents the median. The whiskers extend

to the smallest and largest data points that are not outliers (1.5 times the inter-quartile range). **j** A visualization of top-beta sites in different sleep stages in Montreal Neurological Institute space (wakefulness [$X = -20.7$, $Y = -6.8$, $Z = -5.7$], NREM [$X = -21.5$, $Y = -5.7$, $Z = -5.1$], and REM [$X = -20.7$, $Y = -6.1$, $Z = -5.2$] sleep) relative to the position of internal globus pallidus (GPi) and external globus pallidus (GPe). The three top-beta sites located near, with no significant difference in coordinates in the *X*, *Y*, or *Z* axes ($P = 0.463$, 0.603, and 0.944 for *X*, *Y*, and *Z* axes, respectively, two-sided Kruskal–Wallis test). Three sites of interest are also displayed including (1) the average of all electrodes ($X = -21.4$, $Y = -4.6$, $Z = -3.3$), (2) the mean active contacts ($X = -21.1$, $Y = -5.7$, $Z = -5.3$), and (3) a literature-based coordinate ($X = -22.6$, $Y = -6.7$, $Z = -4.9$) described by Elias et al.[18] to represent the optimal pallidal site for deep brain stimulation in PD. The upper right inset shows the lead localization of all 32 subjects. Source data are provided as a Source Data file.

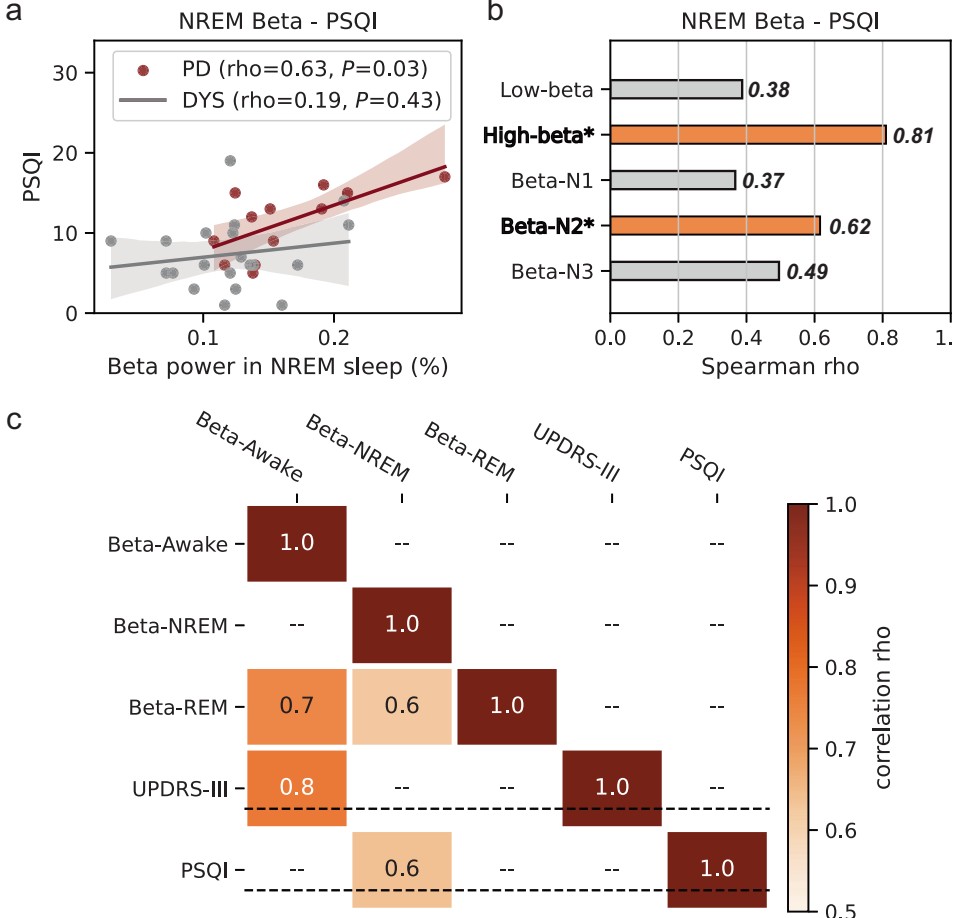

**Fig. 3 | Correlations between pallidal beta power and sleep disturbance ratings in Parkinson's disease. a** Regression plots showing the Spearman correlations between beta power during non-rapid eye movement (NREM) sleep and the Pittsburgh sleep quality index (PSQI) in Parkinson's disease (red) and dystonia (gray). The error bands are the 95% confidence interval for the regression estimate. The null hypothesis is defined as two-sided. **b** Bar plot showing the Spearman correlation coefficients between beta power during NREM sleep and the PSQI when analyzing the high/low beta band and the NREM 1/2/3 stage of sleep. Bars and items

with Spearman correlation *P* < 0.05 are colored in orange and highlighted in bold, respectively. *P* for the two-sided Spearman correlation between NREM high-beta power and PSQI = 0.001; *P* for the two-sided Spearman correlation between beta power in the NREM2 stage and PSQI = 0.033. **c** Heat map showing the correlation matrix between beta power in different sleep–wake stages, off-medication Unified Parkinson's Disease Rating Scale motor score (UPDRS-III), and the PSQI. Squares with Spearman correlation *P* < 0.05 are displayed. Source data are provided as a Source Data file.

## Methods
### Patients and surgery
Thirty-two subjects with movement disorders (12 with PD, age 59.0 [15.0] years, 6 males; 20 with dystonia, age 54.0 [18.0] years, 10 males) scheduled to receive GPi-DBS implantation at Beijing Tiantan Hospital were included. The inclusion criteria were: (i) for primary dystonia patients, predominantly cervical or oromandibular dystonia

without prominent limb involvement; (ii) for PD patients, unquestioned diagnosis of PD based on the UK brain bank criteria, and (iii) for all patients, the capacity to cooperate with whole-night polysomnography recordings and the absence of cerebral lesions on magnetic resonance imaging (MRI) such as tumor and stroke. Standard motor assessments using Toronto Western Spasmodic Torticollis Rating/Burke–Fahn–Marsden Dystonia Rating Scale (for

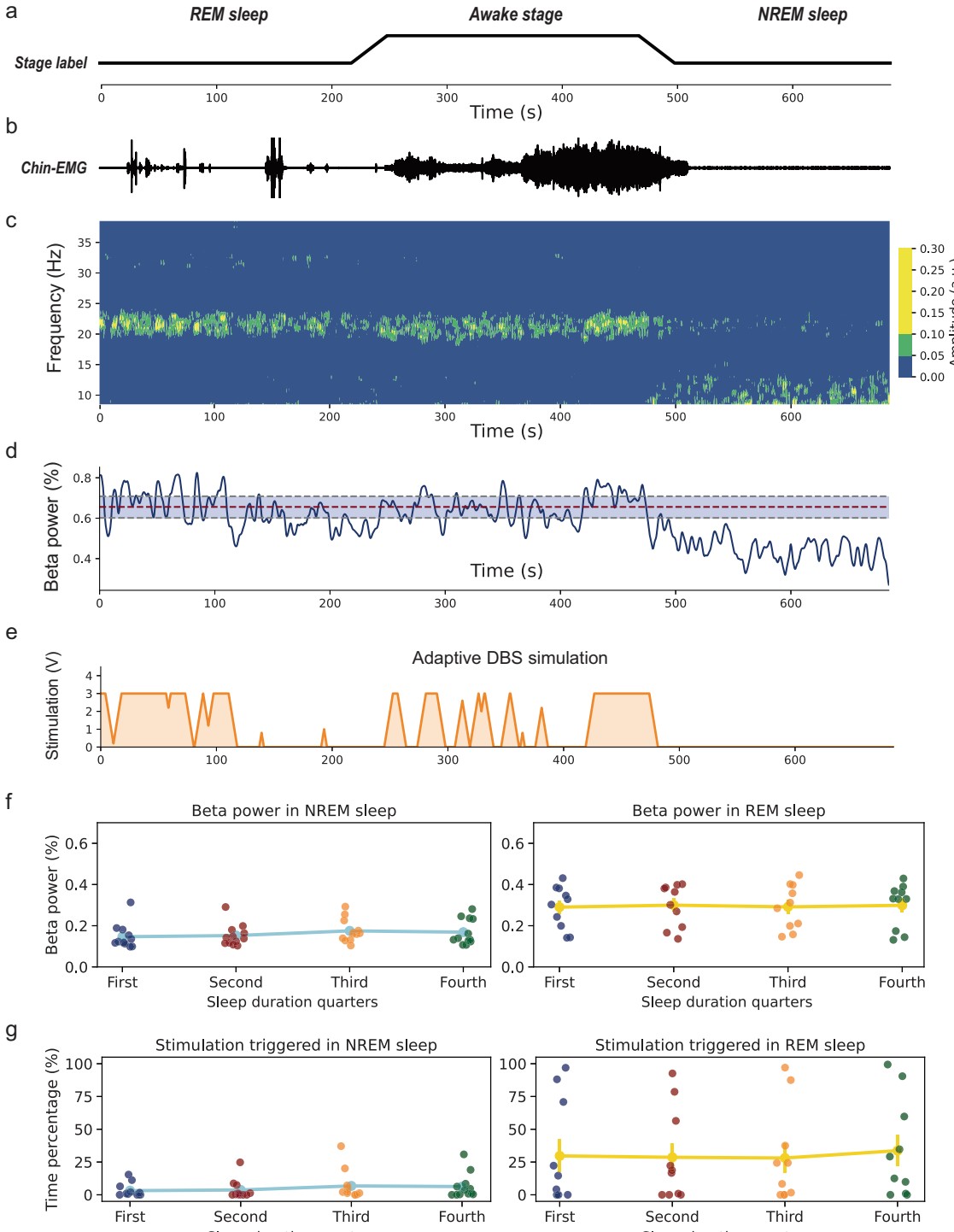

**Fig. 4 | Simulation of adaptive deep brain stimulation during sleep. a** Line plots showing sleep stage definitions of a 700-s recording segment obtained from the subject PD-8. **b** Shows the corresponding chin electromyogram activities of the 700-s recording segment. Note that in wakefulness the electromyogram activities are highest; in rapid eye movement (REM) sleep the background electromyogram activities are lower than that in non-REM (NREM) sleep despite occasional twitch-related electromyogram bursts. **c** Shows the time–frequency representation of pallidal activities of the 700-s sleep recording segment. a.u. refers to the arbitrary units. **d** shows the dynamic beta power change in the 700-s recording segment. The median beta power in wakefulness is labeled with a red dashed line and the 25th and

75th beta power in wakefulness is labeled in gray dashed lines and shaded in light blue. **e** Shows a simulation of adaptive deep brain stimulations using the median beta power in wakefulness as triggering thresholds. **f** Shows the comparison of beta power across four quarters (i.e., 0–25%, 25–50%, 50–75%, and 75–100% of the total length) of NREM (*left*, $P = 0.098$, two-sided Friedman test) and REM sleep length (*right*, $P = 0.782$, two-sided Friedman test). Data are presented as mean values ± SEM. **g** Shows the comparison of ON-stimulation time across four quarters of NREM (*left*, $P = 0.258$, two-sided Friedman test) and REM sleep length (*right*, $P = 0.056$, two-sided Friedman test). Data are presented as mean values ± SEM. Source data are provided as a Source Data file.

dystonia), or MDS-UPDRS-III (for PD, on and off dopaminergic medication) were conducted before the surgery. Subjective sleep quality was evaluated using the PSQI for all subjects and RBD was assessed using the RBD-screening questionnaire for PD subjects. This study conformed to the Declaration of Helsinki and was approved by the local IRB of Beijing Tiantan Hospital. All subjects provided written informed consent. DBS surgery was performed as per routine protocol[44,45]. Guided by the Leksell stereotactic system (Elekta Instrument AB, Stockholm, Sweden), DBS electrodes (model 3387, Medtronic, USA, or model L302, Pins Medical, China) were bilaterally implanted in the GPi region under local anesthesia. Intraoperative electrophysiology recordings using microelectrode and temporary stimulation through microelectrode were conducted for trajectory selection. Precise placement of DBS electrodes was confirmed through postoperative anatomical computed tomography (CT) imaging. Before the second stage of surgery (i.e., the implantation of the pulse generator), DBS leads were externalized for a median of 4 days (range: 3–6 days), during which we recorded whole-night polysomnography and synchronized pallidal local field potentials (hereafter called *sleep recording*).

## Sleep recordings and staging

Sleep recording was conducted during DBS lead externalization, and lasted for one or two successive nights. Data from all recording nights from one subject were pooled together. All anti-dystonia medications were stopped after 12 a.m. (midday) on the day of recording and anti-parkinsonism drugs were stopped after 6 p.m. Post-hoc analysis indicated no residual effects of medication on beta estimation in PD patients (comparison of beta power between four quarters [i.e., the 0–25%, 25–50%, 50–75%, and 75–100% temporal course] of wake after sleep onset [WASO], $Q = 1.32$, $P = 0.724$, two-sided Friedman test). Polysomnography consisting of surface electroencephalogram (EEG), electrooculogram (EOG), and chin electromyogram (EMG) was implemented following standard settings recommended by the American Academy of Sleep Medicine (AASM)[46], as shown in Fig. 1a. Surface bipolar EEG was obtained from frontal (F3-M2, F4-M1), central (C3-M2, C4-M1), and occipital (O1-M2, O2-M1) areas, as per the 10–20-system. Synchronized pallidal LFPs were recorded bipolarly from the adjacent contacts (01, 12, and 23) of each electrode. All signals were amplified ×195, bandpass filtered at 0.08 and 300 Hz and recorded at a sampling rate of 1000 or 2000 Hz through the JE-212 amplifier (Nihon Kohden, Tokyo, Japan). Sleep recording generally started at 9 p.m. (lights out) and ended at 7 a.m. the next morning (lights on). Sleep stages (wake, N1, N2, N3, or REM sleep) were determined in each 30-s epoch by two experienced sleep specialists (N.Z. and J.M.) according to the AASM criteria version 2.6. Importantly, to avoid possible bias induced by the subjectivity of human scorers, all sleep data were also staged using an established open-source sleep staging algorithm (https://github.com/raphaelvallat/yasa)[47], which takes EOG, EEG, EMG, and demographics including age and gender as input. Only epochs with consistent staging results between human and algorithm scorers were qualified for further analysis, accounting for 85.6% of all epochs. Note that patients PD-2/10 and Dyst-10/11/13 had low counts of accurately diagnosed REM sleep epochs ($n < 5$) and were analyzed for NREM sleep and awake states only. A characteristic NREM and REM sleep epoch segment is shown in Fig. 1b and c, respectively. A representative spectrogram from subject PD-8 with the hypnogram on top is shown in Fig. 1d. In the main analysis, N1, N2, and N3 sleep epochs were pooled together to represent NREM sleep while in a supplementary analysis, the three substages of NREM sleep were analyzed separately. In the morning following sleep recording (1–2 h after awakening), we also obtained 5-min artifact-free recordings when the subject was lying in bed in resting wakefulness (hereafter called *awake epochs*).

## Sleep parameters

Based on the polysomnography hypnogram, sleep parameters were extracted including the total time in bed, total sleep time, WASO, sleep latency (time to first sleep epoch), REM sleep latency (time to first REM sleep epoch), the time proportion of N1, N2, N3, and REM sleep, sleep efficiency (total sleep time/total time in bed), sleep fragmentation, and the time proportion of REM sleep without atonia (RSWA). Sleep fragmentation was quantified as the number of events that sleep is interrupted by >2 min of wakefulness. RSWA was determined when the chin EMG (10–70 Hz bandpass) variance in REM sleep was higher than two times the 5th percentile of the chin EMG variance in NREM sleep, which is an established approach listed in the International RBD Study Group (IRBDSG) guidance[48,49]. For a better temporal resolution and as recommended by the IRBDSG[49], the above determination was made for each 3-s REM sleep segment.

## Signal processing and channel selection

Signal analysis was performed using *MNE-Python*[50] and *SciPy*[51]. All signals were notch filtered (Butterworth, bandwidth = 4 Hz, order = 3) to reject the 50 Hz ambient noise and harmonics, and downsampled to 500 Hz. Power spectral density (PSD) of pallidal LFP was computed between 2 and 80 Hz using the Welch periodogram with a fast Fourier transform of 512 points and 50% overlap, which resulted in a frequency resolution of 0.97 Hz. PSD was normalized to the percentage of the total power of 2–45 and 55–80 Hz. Band power of theta (4–12 Hz) and beta (13–30 Hz) was then extracted. In addition, to exclude potential impacts from the aperiodic (1/f-like) component of the signal on our estimation of beta oscillations (the periodic component)[17], we also calculated the aperiodic-adjusted spectrum of the pallidal LFP using the *FOOOF* fitting implemented in Python (https://fooof-tools.github.io/fooof/). We chose the channel located within the GPi to represent pallidal activity, which was confirmed through lead reconstruction (see below). In cases where more than one bipolar channel was within the GPi region, the channel with higher beta during resting wakefulness was selected for analysis for both the PD and dystonia subjects. For cortical signals, we used central EEG channels (i.e., C3 and C4) to represent cortical activities and processed the EEG signals using the same pipeline as we processed pallidal LFPs.

## Beta burst analysis

We conducted the burst dynamic analysis following the criteria that are established by Tinkhauser et al.[20,52]. The only difference was that since very few hemispheres (4/34) in dystonia subjects demonstrated clear beta peaks during NREM or REM sleep, we defined each 1-Hz frequency bin in the beta band range as peak frequency (18 bins, from 13 to 30 Hz) and conducted burst determination based on these individual 18 peak frequencies. The final results were the average of the 18 iterations. Data were first downsampled to 200 Hz and decomposed using Morlet wavelets with 10 cycles. The obtained wavelet amplitude was further z-score normalized to address scale differences. A threshold was defined as the 75th percentile of the amplitude distribution among all hemisphere's data from all subjects. A burst was determined when the instantaneous power crossed the threshold for at least 0.1 s. Burst duration was categorized into six-time windows of 0.15 s starting from 0.1 to >0.85 s in duration. Burst amplitude was defined as the area under the curve between the amplitude curve and the threshold. Burst density was defined as the number of bursts per second.

## Electrode reconstruction and beta power localization analysis

We used the advanced electrode localization pipeline (in Lead-DBS version 2.5.3, MATLAB 2019b) with default settings for electrode reconstruction and group analysis in Montreal Neurological Institute (MNI) space[53]. The pre-operative MRI scan (T1 sequence) was linearly co-registered with the postoperative anatomical CT and nonlinearly

warped to the MNI template (ICBM 2009bNonlinear Asymmetric)[54] using the advanced normalization tools (ANTs)[55]. Electrode trajectories were automatically detected using the PACER approach[56] and were manually refined. The coordinate of a bipolar recording site was defined as the Euclidean midpoint between the two contacts. Sites on the left side were nonlinearly flipped to the right.

To explore the localization of beta power in different sleep stages, we calculated the average coordinate of the recording sites with the top 10% beta power in wakefulness, NREM, and REM sleep. These sites were visualized together with three additional points of interest. First, the average location of all electrodes (regardless of beta). Second, the active DBS contacts are used in optimal programming. Third, a literature-based coordinate described by Elias et al.[18] to represent the optimal pallidal site for DBS in PD.

### Adaptive DBS simulation analysis

To explore how adaptive DBS would deliver stimulation in a real context of nocturnal sleep, we segmented a continuous 700-s data in an exemplary PD subject (PD-8) that included all three stages of wakefulness, NREM, and REM sleep. We calculated beta power using the aforementioned Welch periodogram method with a sliding window of 5 s and a step of 0.5 s. The obtained time-resolved beta activity was smoothed using a Gaussian filter (sigma = 5) before being input to the threshold-based adaptive DBS control algorithms. We simulated adaptive DBS stimulation using a common threshold-based strategy. The stimulation-triggering threshold was set as the 50th beta power of wakefulness. The lower and upper stimulation voltage border was 0 and 3 V, respectively. The ramping speed was simulated to be 0.4 V/s. At last, we calculated the stimulation-on time under the individual threshold of median beta power of wakefulness for all PD subjects. Stimulation-on time was compared between NREM and REM sleep and between four sleep quarters (i.e., 0–25%, 25–50%, 50–75%, and 75–100% of the total length) to test potential influences of early vs. late stages of sleep on adaptive DBS algorithms over the course of the night.

### Statistical analysis

We performed all statistical tests using *NumPy*[57] and *SciPy*[51] in Python 3.8. Given that most studied variables were of non-normal distribution, data were presented as the median (interquartile range). Nonparametric tests including the Mann–Whitney $U$ test, Kruskal–Wallis test, and Friedman test were used and indicated when used. We employed the Spearman correlation to test potential correlative relationships between beta power in different stages and clinical scales of interest. Multiple comparisons were controlled using the Bonferroni correction with the $P$ value marked as $P_{Bonferroni}$. Two-tailed $P$-values < 0.05 were considered significant.

### Reporting summary

Further information on research design is available in the Nature Portfolio Reporting Summary linked to this article.

## Data availability

Due to the data protection regulations of Beijing Tiantan Hospital and Medical Capital University, the raw sleep electrophysiological data used in this study are available from the corresponding author (J.G.Z.) after approval of the IRB of Beijing Tiantan Hospital (E-mail: ttyyirb@163.com, Tel: +86 10 5997 8555). We are happy to share our data and provide assistance in obtaining the approval upon request. Source data are provided with this paper.

## Code availability

All relevant codes employed in the study can be freely accessed without restriction at https://github.com/zixiao-yin/SleepBeta (Zenodo https://doi.org/10.5281/zenodo.8180780[58]).

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

## Acknowledgements

J.G.Z. is supported by the National Nature Science Foundation of China (81830033, 61761166004). W.-J.N. receives funding from the European Union (ERC, ReinforceBG, project 101077060), Deutsche Forschungsgemeinschaft (DFG, German Research Foundation)—Project-ID 424778381—TRR 295 ReTune and the Bundesministerium für Bildung und Forschung (BMBF, project FKZ01GQ1802). A.C.Y. receives funding from the National Nature Science Foundation of China (81870888). G.Y.Z. receives funding from the National Nature Science Foundation of China (82201634). N.Z. receives funding from the National Key Research and Development Program of China (2020YFC2005300). A.A.K. received funding from the Lundbeck Foundation project grant "Adaptive and precise targeting of cortex-basal ganglia circuits in Parkinson's Disease" (Grant No. R336-2020-1035). A.A.K. and H.B. receive funding from the Deutsche Forschungsgemeinschaft (DFG, German Research Foundation)—Project-ID 424778381—TRR 295 ReTune.

## Author contributions

Z.X.Y., W.-J.N., and Z.J.G. designed the research; Z.X.Y., R.Y.M., Q.A., Y.C.X., Y.F.G., G.Y.Z., Y.J., N.Z., Y.A.C., and M.F.G. acquired the data; Z.X.Y., W.-J.N., and Z.J.G. made the analysis and interpretation of data; Z.X.Y. made the first draft of manuscript; Z.X.Y., W.-J.N., A.A.K., H.B., and Z.J.G. performed the revision of manuscript.

## Competing interests

W.-J.N. received honoraria for talks unrelated to this manuscript from Medtronic which is a manufacturer of deep brain stimulation devices. A.A.K. is on the advisory board of Boston Scientific and Medtronic and has received honoraria unrelated to this manuscript from Boston Scientific, Medtronic, Stadapharm, and Teva, companies manufacturing deep brain stimulation or pharmaceutical therapies. H.B. is a consultant of Alpha Omega and has received travel honoraria from Boston Scientific and Medtronic unrelated to this manuscript. The remaining authors declare no competing interests.
