## [Peer Review File · Nature Communications]

Pathological pallidal beta activity in Parkinson's disease is sustained during sleep and associated with sleep disturbanceREVIEWER COMMENTS

Reviewer #1 (Remarks to the Author):

In this paper, the authors study the relationship between the beta activity recorded in the globus pallidus, both during wakefulness and in different stages of sleep, and sleep disturbances in Parkinson's disease, which are frequent and have great impact on the quality of life of these patients. As novelty, the authors use a group of patients with dystonia as a control. In a second step, they apply this knowledge to estimate how demand-dependent adaptive stimulation would perform during sleep with parameters calculated while awake. They find that in patients with Parkinson's disease, and not in patients with dystonia, there is a correlation between beta activity recorded during sleep (especially in the high beta range) and sleep disorders. The correlation is maximum when the NREM2 stage of sleep is analyzed and it does not exist analyzing beta activity during wakefulness. Interestingly, the authors estimate that the dynamics of beta activity throughout wakefulness and the different stages of sleep imply that demand-dependent adaptive stimulation as usually programmed would fail during sleep, thus probably not improving sleep disturbances.

From my point of view, the method is very meticulous and the findings are interesting and clinically useful.

That being said, I have some comments.

- In my opinion, the title could be improved. I think that it should specify that it is beta pallidal activity. I understand that the length of the title may be a limitation, but ideally, it should include a reference to the stimulation part.
- Figure legends, especially those of supplementary information, are very long and contain many results that I think would be better placed in the main body of the article.
- Figure 2f: the text of the legend is not readable due to the background.
- Figure 3a and 3c: the graph legend overlaps the y-axis
- Figure 4f and 4g: I would appreciate an explanation of what the authors mean by sleep quantiles.
- Table 1: it is not well configured and it is difficult to identify which values correspond to "median (IQR)"

Reviewer #2 (Remarks to the Author):

In this cooperation paper with patients from Beijing China, and contribution of authors from Germany and Israel, the authors attempted to investigate the clinical potential of beta activity as biomarker for sleep quality in PD. They recorded pallidal activity invasively during PSG in PD patients and dystonia patients as controls. They found increased beta activity across wakefulness, REM and non REM sleep, which was correlated with sleep disturbance. The manuscript is relevant with respect to implement sleep adaptive DBS. The authors state that excessive beta activity, hallmark of hypodopaminergic Parkinson could be used as biomarker for adaptive DBS.

The study design using dystonia patients as controls because they also have DBS, is well selected. The sample size is still very small, 12 PD and 20 dystonia had recordings of beta-activity during sleep combined with regular polysomnography. Sleep analysis was done by algorithm and visual analysis, and classical sleep analysis confirmed results from earlier studies about distinct findings in PD. In addition, in the morning after sleep recording, a five minute recording of resting wakefulness was obtained. The authors found that pallidal beta activity in PD was significantly higher during nonREM and REM sleep in PD compared to controls. The study also has very nice and well done graphs.

Later, they correlated the pallidal beta-power during non REM and REM episodes with sleep quality and RBD ratings. For sleep quality they used the PSQI, an old, but very much used and very well validated scale for sleep quality. And, they found that pallidal beta power was correlated with RBD SQ in non REM, but not in REM sleep. Variability was also correlated with pallidal beta activity in awake stage.

For the reviewer it seems difficult to explain, how the RBD screening questionnaire, which has limited sensitivity and specificity even in PD patients, and in light of the significant drawbacks of using RBD screening questionnaires instead of good clinical history and video polysomnographic analysis to diagnose RBD according to ICSD3 clinical and IRBDSG research criteria. The RBD SQ, which gives higher score to subjects diagnose with PD, seems questionable in this context, and the higher IRBDSQ Score could potentially indicate general or confounder sleep disturbance, not RBD severity or i confounding influences. The RBDSQ does not say anything about the severity of RBD. Therefore, this finding is the weakest one in the whole paper.

In general, this manuscript is innovative, very well written, and presents interesting new

findings. The only thing that might perhaps better be taken out, or would need to be discussed much more critically is the relation with the RBDSQ.

Reviewer #3 (Remarks to the Author):

The authors present a well-conducted and interesting study assessing pallidal activity (invasive recording in patients with DBS) during sleep in patients with Parkinson's disease and dystonia. In the Parkinson group, beta activity was elevated not only during wakefulness but also during sleep (REM and non-REM), and correlated with subjectively reported sleep disturbances (as assessed by the Pittsburgh Sleep Quality Index). To investigate the role of adaptive stimulation during sleep, the authors performed a simulation of adaptive stimulation, showing that sleep related beta activity changes remain unaccounted for by currently used algorithms, potentially leading to a negative outcome in sleep quality. Based on these data, the authors suggest that future studies should assess improvement in adaptive stimulation algorithms to improve sleep disturbance treatment in Parkinson's disease.

The study is novel and relevant for the field. I have some comments and suggestions.

Major issues:

1. The authors report that in 2 patients with PD and 3 with dystonia only less than 5 REM sleep epochs were accurately diagnosed. Was this the case in all nights? If so, what was the cause of this issue? If the problem did not affect all nights, REM sleep during scorable nights should be included. In both cases, this issue should be listed in the limitations.
2. Fig. 2 d, e, f shows a relevant overlap in power spectra between the Parkinson and the dystonia group during NREM sleep. This should be discussed.
3. REM sleep without atonia values should be reported, as well as the fact that RBD was not confirmed by PSG in any of the PD patients (despite some were above the RBDSQ cut-off). Correlation between REM sleep without atonia and pallidal beta power should be reported. This should then be discussed, instead of the correlation with the RBDSQ (which has been shown to have low specificity for RBD). Also, in the discussion part about RBD, the authors should consider that RBD may not relate to more severe motor symptoms, but to a more severe PD phenotype (with autonomic dysfunction, dementia, ...).

Minor:

1. Findings and conclusions should not be mentioned in the introduction. I would suggest to end the introduction with the aims. Also, methods are reported in the first part of the results section and should be moved to the methods section.
2. I would suggest a language review for the discussion.
3. In the last sentence of the limitation, I would suggest using "basal ganglia during sleep" instead of "sleeping basal ganglia", to avoid misinterpretations.

Reviewer #1

1. In this paper, the authors study the relationship between the beta activity recorded in the globus pallidus, both during wakefulness and in different stages of sleep, and sleep disturbances in Parkinson's disease, which are frequent and have great impact on the quality of life of these patients. As novelty, the authors use a group of patients with dystonia as a control. In a second step, they apply this knowledge to estimate how demand-dependent adaptive stimulation would perform during sleep with parameters calculated while awake. They find that in patients with Parkinson's disease, and not in patients with dystonia, there is a correlation between beta activity recorded during sleep (especially in the high beta range) and sleep disorders. The correlation is maximum when the NREM2 stage of sleep is analyzed and it does not exist analyzing beta activity during wakefulness. Interestingly, the authors estimate that the dynamics of beta activity throughout wakefulness and the different stages of sleep imply that demand-dependent adaptive stimulation as usually programmed would fail during sleep, thus probably not improving sleep disturbances. From my point of view, the method is very meticulous and the findings are interesting and clinically useful.

Reply:

We thank the reviewer's positive feedback!

2. That being said, I have some comments.

- In my opinion, the title could be improved. I think that it should specify that it is beta pallidal activity. I understand that the length of the title may be a limitation, but ideally, it should include a reference to the stimulation part.

Reply:

We agree with the reviewer. We modified our title which now specifies that it is pallidal beta activity and better covers the stimulation part. The title now reads:

“Parkinson’s sleep-related exaggerated pallidal beta activity remains unaccounted for by current adaptive DBS algorithms.”

We hope this could be a better title and thank the reviewer’s constructive suggestion.

3. - Figure legends, especially those of supplementary information, are very long and contain many results that I think would be better placed in the main body of the article.

Reply:

We thank the reviewer’s suggestion. Two aspects of efforts are made to address this problem. First, some results or statistics reported in the supplementary information are now moved to the main text. For example:

(1) The specific statistics in the comparison of beta power between PD and dystonia groups in N1, N2, and N3 sleep.

“Substage analysis of NREM sleep indicated that beta power was higher in patients with PD in N1 ($P = 0.006$) and N2 ($P = 0.013$), but not in N3 sleep ($P = 0.093$, Mann–Whitney U test) than patients with dystonia (**Supplementary Fig. 1**).”

(2) Correlation results for the high/low beta and the substages of NREM sleep.

“Pallidal beta power during NREM sleep was robustly correlated with the Pittsburgh sleep quality index (PSQI) in PD (Spearman $\rho = 0.63$, $P = 0.028$) but not dystonia patients (Spearman $\rho = 0.19$, $P = 0.434$, **Fig. 3a**). This correlation was more robust for high beta (20-30 Hz) than low beta (13-20 Hz) power, and for NREM2 than NREM1/3 stages of sleep (Fig. 3b).”

Fig. 3 Correlations between pallidal beta power and sleep disturbance ratings in Parkinson's disease. **a** Regression plots showing the Spearman correlations between beta power during non-rapid eye movement (NREM) sleep and the Pittsburgh sleep quality index (PSQI) in Parkinson's disease (red) and dystonia (gray). **b** Bar plot showing the Spearman correlation coefficients between beta power during NREM sleep and the PSQI when analyzing the high/low beta band and the NREM 1/2/3 stage of sleep. Bars with Spearman correlation $P < 0.05$ are colored in orange. **c** Heat map showing the correlation matrix between beta power in different sleep-wake stages, off-medication Unified Parkinson's Disease Rating Scale motor score (UPDRS-III), and the PSQI. Squares with Spearman correlation $P < 0.05$ are displayed.

Second, legends in the supplementary information are shortened to increase readability. For example, the original figure legend in the Supplementary Figure 1 is "Beta power is significantly higher in Parkinson's disease than dystonia in N1 (dystonia vs. Parkinson's disease: 0.138 [0.116–0.187] vs. 0.255 [0.179–0.343], ** $P = 0.006$, Mann-Whitney U test) and N2 (dystonia vs. Parkinson's disease: 0.119 [0.098–0.137] vs.

0.157 [0.123–0.193], * $P = 0.013$, Mann–Whitney U test), but not N3 sleep (dystonia vs. Parkinson’s disease: 0.083 [0.066–0.124] vs. 0.106 [0.091–0.155], $P = 0.093$, Mann–Whitney U test).”

Now it reads:

“Beta power is significantly higher in Parkinson’s disease than dystonia in N1 (** $P = 0.006$, Mann–Whitney U test) and N2 (* $P = 0.013$, Mann–Whitney U test), but not N3 sleep ($P = 0.093$, Mann–Whitney U test).”

4. - Figure 2f: the text of the legend is not readable due to the background.

Reply:

We change the color of background and add a semitransparent light white box behind the legend text. Now the text is better readable. We thank the reviewer’s suggestion.

5. - Figure 3a and 3c: the graph legend overlaps the y-axis

Reply:

This has been revised. Please see below.

6. - Figure 4f and 4g: I would appreciate an explanation of what the authors mean by sleep quantiles.

Reply:

We apologize for the lack of clarity here. By mentioning sleep quantiles, we intended to describe the chronological order of sleep time points from onset to end adapted to the total length of sleep (e.g., if the patient slept for 6 hours, the quantiles would describe 1.5h / 90 minutes from onset to end). The intention of this analysis was to potentially uncover differences in the development of beta power over the course of the night while adjusting to differences in total sleep times. We have rephrased the original statement, which now reads as follows:

In the legend of Figure 4,

“f shows the comparison of beta power across four quarters (i.e., 0-25%, 25-50%, 50-75%, and 75-100% of the total length) of NREM (left, $P = 0.098$, Friedman test) and REM sleep length (right, $P = 0.782$, Friedman test).”

In the Methods section,

“Stimulation-ON time was compared between NREM and REM sleep and between four sleep quarters (i.e., 0-25%, 25-50%, 50-75%, and 75-100% of the total length) to test potential influences of early vs. late stages of sleep on adaptive DBS algorithms over the course of the night.”

7. - Table 1: it is not well configured and it is difficult to identify which values correspond to “median (IQR)”

Reply:

We thank the reviewer’s great suggestion. Table 1 has now been re-configured. Please see below. We hope that the journal typesetting will further help with an optimized design.

Table 1 Demographics and clinical characteristics of the included patients

Patient	Nights (n)	Sex	Age (y)	DD (y)	PSQI	RBDSQ	Motor score ^b	Medication
Parkinson’s disease								
PD-1	1	f	72	20	16	11	46/28	Madopar, Amantadine, Piribedil
PD-2	1	m	52	7	15	10	26/11	Madopar, Rasagiline
PD-3	1	m	42	6	9	2	53/23	Madopar, Sinemet, Pramipexole
PD-4	1	f	67	6	13	5	33/21	Madopar, Piribedil, Entacapone, Pramipexole
PD-5	1	m	48	5	6	1	29/15	Madopar
PD-6	1	m	52	8	9	1	45/31	Madopar, Piribedil
PD-7	2	f	60	10	13	7	69/31	Madopar, Sinemet
PD-8	2	f	58	6	6	1	71/32	Sinemet, Piribedil, Entacapone
PD-9	2	m	72	12	15	2	54/18	Madopar, Sinemet, Amantadine
PD-10	1	f	66	9	12	12	67/32	Madopar, Trihexyphenidy, Piribedil
PD-11	1	f	59	7	5	1	45/23	Madopar
PD-12	2	m	59	8	17	13	66/37	Madopar, Entacapone, Pramipexole
Median (IQR)			59.0 (15.0)	7.5 (4.0)	15.0 (7.0)	3.5 (9.5)	49.5 (30.8)/ 25.5 (13)	
Dystonia								
Dyst-1	2	f	50	2	5	-	C/20	Benzhexol, Baclofen, Clonazepam

Dyst-2	2	m	65	3	10	-	M/8	Clonazepam, Tiapride hydrochloride, Mecobalamin
Dyst-3	1	m	53	5	9	-	M/7	Botulin, Carbamazepine
Dyst-4	2	m	59	5	9	-	M/23	Clonazepam
Dyst-5	1	f	64	4	6	-	M/6	Baclofen
Dyst-6	1	f	21	4	1	-	C/28	Amantadine, Benzhexol
Dyst-7	1	m	65	3	19	-	M/12	Tiapride hydrochloride
Dyst-8	1	f	60	5	14	-	M/13	Tiapride hydrochloride
Dyst-9	1	f	58	7	5	-	M/16	Botulin
Dyst-10	2	m	45	3	3	-	C/32	NA
Dyst-11	1	m	65	15	11	-	M/18	Botulin
Dyst-12	1	f	55	15	1	-	M/7	NA
Dyst-13	1	f	30	4	5	-	C/22	Baclofen, Clonazepam
Dyst-14	1	f	59	3	3	-	M/9	Clonazepam
Dyst-15	1	m	43	5	11	-	M/15	Benzhexol, Haloperidol
Dyst-16	1	m	42	6	7	-	C/38	NA
Dyst-17	1	f	45	7	6	-	M/13	Botulin, Tiapride hydrochloride
Dyst-18	1	f	59	3	6	-	M/16	Botulin
Dyst-19	1	m	29	6	10	-	C/24	Clonazepam
Dyst-20	1	m	17	2	6	-	C/17	Baclofen, Haloperidol
Median (IQR)			54.0 (18.0)	4.5 (3.0)	6.0 (5.0)	-		

DD, duration of disease; PSQI, Pittsburgh sleep quality index; RBDSQ, REM sleep behavior disorder-screening questionnaire; Dyst, dystonia; PD, Parkinson's disease; f, female; m, male; C, cervical dystonia; M, Meige syndrome (oromandibular dystonia); NA, not applicable.

^aSleep scales were the Pittsburgh sleep quality index for dystonia, and the Pittsburgh sleep quality index/ REM Sleep Behavior Disorder Screening Questionnaire for Parkinson's disease.

^bPreoperative motor score was the Toronto Western Spasmodic Torticollis Rating for cervical dystonia, Burke–Fahn–Marsden Dystonia Rating Scale (movement) for oromandibular dystonia, and MDS-Unified Parkinson's Disease Rating Scale-III off/on medication for Parkinson's disease.

Reviewer #2

1. In this cooperation paper with patients from Beijing China, and contribution of authors from Germany and Israel, the authors attempted to investigate the clinical potential of beta activity as a biomarker for sleep quality in PD. They recorded pallidal activity invasively during PSG in PD patients and dystonia patients as controls. They found increased beta activity across wakefulness, REM and non REM sleep, which was correlated with sleep disturbance. The manuscript is relevant with respect to implement sleep adaptive DBS. The authors state that excessive beta activity, hallmark of hypodopaminergic Parkinson could be used as biomarker for adaptive DBS.

The study design using dystonia patients as controls because they also have DBS, is well selected. The sample size is still very small, 12 PD and 20 dystonia had recordings of beta-activity during sleep combined with regular polysomnography. Sleep analysis was done by algorithm and visual analysis, and classical sleep analysis confirmed results from earlier studies about distinct findings in PD. In addition, in the morning after sleep recording, a five minute recording of resting wakefulness was obtained. The authors found that pallidal beta activity in PD was significantly higher during nonREM and REM sleep in PD compared to controls. The study also has very nice and well done graphs. Later, they correlated the pallidal beta-power during non REM and REM episodes with sleep quality and RBD ratings. For sleep quality they used the PSQI, an old, but very much used and very well validated scale for sleep quality. And, they found that pallidal beta power was correlated with RBD SQ in non REM, but not in REM sleep. Variety was also correlated with pallidal beta activity in awake stage.

Reply:

We thank the reviewer's recognition on our work.

2. For the reviewer it seems difficult to explain, how the RBD screening

questionnaire, which has limited sensitivity and specificity even in PD patients, and in light of the significant drawbacks of using RBD screening questionnaires instead of good clinical history and video polysomnographic analysis to diagnose RBD according to ICSD3 clinical and IRBDSG research criteria. The RBD SQ, which gives higher score to subjects diagnose with PD, seems questionable in this context, and the higher IRBDSQ Score could potentially indicate general or confounder sleep disturbance, not RBD severity or confounding influences. The RBDSQ does not say anything about the severity of RBD. Therefore, this finding is the weakest one in the whole paper.

In general, this manuscript is innovative, very well written, and presents interesting new findings. The only thing that might perhaps better be taken out, or would need to be discussed much more critically is the relation with the RBDSQ.

Reply:

We thank the reviewer for pointing this out. We critically reflected on the presentation of our results and fully agree with the reviewer that the RBDSQ is not a scale that tracks the severity of RBD. We also agree that the higher RBDSQ score could potentially indicate generally more severe sleep disturbances that could be unspecific. In our cohort, there was a strong correlation between PSQI and RBDSQ score (Spearman $\rho = 0.821$, $P = 0.001$). But before taking this section of results out, we made some more analysis to further study the relation between beta power during NREM sleep and the severity of RBD.

In the quest to try and find a more objective physiological proxy of RBD severity also in reply to reviewer #3, we calculated the proportion of REM sleep time without atonia (RSWA). The RSWA is determined when chin EMG (10-70 Hz bandpass) variance during REM sleep was higher than two times the 5th percentile of the chin EMG variance during NREM sleep¹. We then correlated the beta power during NREM sleep with the severity of RSWA and found that the correlation is weakly positive but insignificant (Spearman $\rho = 0.26$, $P = 0.47$), but the severity of RSWA is also only

moderately correlated with RBDSQ and insignificant (Spearman $\rho = 0.55$, $P = 0.09$). Therefore, as suggested by the reviewer, we take these inconclusive results out from the main text to the supplementary information and discussed the relation more critically in the discussion section.

In the Results section,

“Pallidal beta power during NREM sleep was robustly correlated with the Pittsburgh sleep quality index (PSQI) in PD (Spearman $\rho = 0.63$, $P = 0.028$) but not dystonia patients (Spearman $\rho = 0.19$, $P = 0.434$, **Fig. 3a**). This correlation was more robust for high beta (20-30 Hz) than low beta (13-20 Hz) power, and for NREM2 than NREM1/3 stages of sleep (**Fig. 3b**). A significant correlation was identified between beta power during NREM sleep and the RBD-Screening Questionnaire (RBDSQ) score, but this finding may have limited specificity as no correlation was found when analyzing the proportion of REM sleep time without atonia as a proxy of RBD severity (Supplementary Fig. 5).”

Supplementary Fig. 5 Correlations between pallidal beta band power and REM sleep behavior

disorder severity ratings. The heatmap on the right shows that beta power in NREM (Spearman $\rho = 0.67$, $P = 0.02$), but not REM sleep is significantly correlated with the REM sleep behavior disorder-screening questionnaire (RBDSQ) score. However, no correlation is observed between beta power in NREM sleep and the time proportion/severity of REM sleep without atonia (RSWA) (Spearman $\rho = 0.26$, $P = 0.47$), despite that the severity of RSWA is moderately correlated with the RBDSQ score

(Spearman $\rho = 0.55$, $P = 0.09$). Scatter plots for the abovementioned correlations are shown separately on the left.

In the Discussion section,

“In the present cohort, dream enactment was not observed in any of the included PD patients, but NREM but not REM related beta activity was found to be associated with higher scores in the RBD-Screening Questionnaire. Given that this questionnaire is unspecific and does not provide a continuous scaling of RBD severity, we further quantified the loss of REM sleep atonia, as an indirect marker of RBD severity. We observed REM sleep without atonia in a median of 26.1% of all REM sleep epochs in PD patients, but found no significant correlation of the relative proportion of these epochs with beta activity measures. Thus, considering the limited sensitivity and specificity of the RBD-Screening Questionnaire in diagnosing REM sleep behavior disorder²⁹, we speculate that the correlation between NREM beta power and RBDSQ could rather corroborate our abovementioned finding that beta can generally index sleep disturbances, which is further substantiated by the high correlation between PSQI and RBDSQ scores ($\rho = 0.821$, $P = 0.001$).”

In the Methods section, a new paragraph is added to describe how the RSWA and other sleep parameters are calculated.

“*Sleep parameters*

Based on the polysomnography hypnogram, sleep parameters were extracted including the total time in bed, total sleep time, WASO, sleep latency (time to first sleep epoch), REM sleep latency (time to first REM sleep epoch), the time proportion of N1, N2, N3, and REM sleep, sleep efficiency (total sleep time/total time in bed), sleep fragmentation, and the time proportion of REM sleep without atonia (RSWA). Sleep fragmentation was quantified as the number of events that sleep is interrupted by > 2 minutes of wakefulness. RSWA was determined when the chin EMG (10-70 Hz bandpass) variance in REM sleep was higher than two times the 5th percentile of

the chin EMG variance in NREM sleep⁴⁸. For a better temporal resolution and as suggested by the International RBD Study Group⁴⁹, the above determination was made for each 3-second REM sleep segment.”

The proportion of REM sleep time without atonia is also reported in Table 2. Please see below.

Table 2 Sleep parameters of the included patients^a

	Parkinson’s disease (n = 12)	Dystonia (n = 20)	P values ^b
TTB (min)	494.0 (51.2)	509.0 (59.0)	0.599
TST (min)	298.4 (108.2)	368.2 (75.0)	0.012
WASO (min)	222.0 (143.0)	138.4 (55.0)	0.129
SL (min)	50.1 (55.7)	14.2 (23.7)	0.049
RSL (min)	216.4 (137.6)	101.2 (70.0)	0.008
N1pct (%)	8.5 (14.9)	6.3 (6.7)	0.712
N2pct (%)	60.6 (15.4)	64.4 (12.6)	0.448
N3pct (%)	14.9 (10.9)	8.1 (10.8)	0.134
Rpct (%)	10.7 (7.0)	17.4 (9.8)	0.006
SE (%)	59.9 (29.6)	74.3 (12.9)	0.064
Sfrag (n) ^c	15.5 (8.8)	9.5 (5.2)	0.015
RSWA (%) ^d	26.1 (24.9)	7.8 (14.8)	0.083

TTB, total time in bed; TST, total sleep time; WASO, wake after sleep onset; SL, sleep latency; RSL, REM sleep latency; N1pct, percentage of the N1 sleep; N2pct, percentage of the N2 sleep; N3pct, percentage of the N3 sleep; Rpct, percentage of the REM sleep; SE, sleep efficiency; Sfrag, sleep fragmentations; RSWA, REM sleep without atonia.

^aDescriptive data are presented as the median (interquartile range).

^b*P* Statistics are obtained using Mann–Whitney *U* test without applying the Bonferroni correction. Significant comparisons with uncorrected *P* value < 0.05 are highlighted in bold.

^cSleep fragmentation is quantified as the times that sleep is interrupted by > 2 minutes’ wakefulness.

^dRSWA is quantified as the percentage of REM sleep time when EMG activities are higher than two

times the 5th percentile of the chin EMG activities during NREM sleep (detailed in Materials and Methods).

Reviewer #3

1. The authors present a well-conducted and interesting study assessing pallidal activity (invasive recording in patients with DBS) during sleep in patients with Parkinson's disease and dystonia. In the Parkinson group, beta activity was elevated not only during wakefulness but also during sleep (REM and non-REM), and correlated with subjectively reported sleep disturbances (as assessed by the Pittsburgh Sleep Quality Index). To investigate the role of adaptive stimulation during sleep, the authors performed a simulation of adaptive stimulation, showing that sleep related beta activity changes remain unaccounted for by currently used algorithms, potentially leading to a negative outcome in sleep quality. Based on these data, the authors suggest that future studies should assess improvement in adaptive stimulation algorithms to improve sleep disturbance treatment in Parkinson's disease. The study is novel and relevant for the field.

Reply:

We thank the reviewer's positive feedback.

2. I have some comments and suggestions.

Major issues:

The authors report that in 2 patients with PD and 3 with dystonia only less than 5 REM sleep epochs were accurately diagnosed. Was this the case in all nights? If so, what was the cause of this issue? If the problem did not affect all nights, REM sleep during scorable nights should be included. In both cases, this issue should be listed in the limitations.

Reply:

We thank the reviewer's comments and apologize for the lack of clarity here. The main cause for the low count of REM sleep is that the 5 patients suffered from very poor sleep quality, e.g., subjects PD-2, PD-10, and Dyst-10 had a sleep efficiency of 47.1%,

27.4%, and 56.4%, respectively. This may have led to the fact that REM sleep was severely disturbed in these patients. When we performed a literature review, we found similar reports of reduced or even abolished REM sleep in patients with neurological disorders^{2,3}. For the 5 abovementioned patients (PD-2/10, Dyst-10/11/13), the time that is scored as potential REM sleep by clinical sleep specialists is 0, 8, 11.5, 9.5, and 26 minutes, respectively, significantly lower than the average of around 60 minutes of REM sleep in all patients. In addition, as clarified in the Methods section, to get a stage scoring result that is as unbiased and accurate as possible, we employed a two-step staging approach consisting of manual and algorithm-based staging. After the algorithm staging, the count of epochs that were deemed as accurately diagnosed REM sleep was less than 5 in all patients. Here are some additional remarks:

Regarding the “scorable nights” suggested by the reviewer, it is disappointing that only one patient (Dyst-10) had two nights of recording (see Table 1) and the count of REM sleep epochs on the other night for that patient is even lower (2.5 minutes staged by the human scorer and 1 epoch after algorithm checking), rendering it not analyzable.

Regarding the staging discrepancy between human and algorithm scorers, we speculate that human scorers are more inclined to label uncertain REM epochs as REM sleep especially when they found the patient has little or no REM sleep, which is rare for a normal night of sleep. The two-step verification approach, though admittedly reduces the amount of data that can be analyzed, could be helpful in alleviating the potential influence of subjectivity in sleep scoring⁴ and reducing the possibility that uncertain sleep epochs contaminate either stage group. Multi-night recordings in an in-home environment could be a promising improvement in obtaining more data from patients with poor sleep quality but are unavailable in this setting of acute invasive brain signal recordings. In the future, such data could be obtained through sensing-enabled brain stimulation implants.

As suggested by the reviewer, this point has now been added to the limitations.

“We would like to highlight the following limitations of our study. First, even though the number of included subjects in this study is higher than most previous

studies with similar aims^{14,29} the sample size is still relatively small. Second, subjects suffering from poor sleep quality had reduced or even completely abolished REM sleep^{14,16}, which further reduces the amount of data for REM sleep analyses in this study. Recordings across multiple nights could be a potential solution to this problem.”

3. Fig. 2 d, e, f shows a relevant overlap in power spectra between the Parkinson and the dystonia group during NREM sleep. This should be discussed.

Reply:

We thank the reviewer’s suggestion. The relevant overlap in power spectra between the PD and dystonia group during NREM sleep could be due to that beta activities in NREM sleep are dramatically reduced in PD compared to REM sleep and wakefulness. But since the variance of data is also reduced, beta power in PD is still significantly higher than that in dystonia, which can be corroborated by the further beta burst and FOOOF analysis.

From the substage analysis of NREM sleep, we noticed that for both the Parkinsonian and dystonia patients, beta power reduced from N1 to N2 to N3 sleep, and in N3 sleep beta power is comparable between PD and dystonia patients. We suggested a potential normalization effect exerted by slow wave sleep on basal ganglia oscillations here and suggested future studies to investigate whether suppression of basal ganglia beta activities could inversely result in longer deep sleep in PD.

These discussions are now added to the discussion section.

“As previously reported, we found a drop in beta activity during NREM stages, when compared to REM and wakefulness, resulting in a relative overlap of power spectra across the PD and the dystonia groups. However, despite the amplitude decrease, our results suggest that beta power remains excessively high during NREM sleep, when comparing PD with dystonia subjects. In addition, we found evidence that this pathophysiological pattern during the NREM phase may be

associated with impaired sleep quality, as beta power during NREM sleep showed robust correlations with the PSQI, a validated assessment of sleep disturbance. This correlation was most robust in the NREM2 stage of sleep, potentially because N2 sleep occupied the longest duration of sleep time (over 60% in our patient cohort) and that physiologically important sleep oscillations such as spindles and K-complexes are typically most prominent in N2 sleep²⁶. When comparing beta power across sleep cycles in the PD cohort, we found a relative decline in beta power from N1 sleep, where it was close to that in wakefulness, to N3 sleep, where it was more similar to that in dystonia. Whether this indicates a potential “normalization” effect of N3 sleep and whether, inversely, a suppression of basal ganglia beta activities could result in longer deep sleep in PD requires further investigation.”

4. REM sleep without atonia values should be reported, as well as the fact that RBD was not confirmed by PSG in any of the PD patients (despite some were above the RBDSQ cut-off). Correlation between REM sleep without atonia and pallidal beta power should be reported. This should then be discussed, instead of the correlation with the RBDSQ (which has been shown to have low specificity for RBD). Also, in the discussion part about RBD, the authors should consider that RBD may not relate to more severe motor symptoms, but to a more severe PD phenotype (with autonomic dysfunction, dementia, ...).

Reply:

We thank the reviewer’s constructive suggestion. Following the suggestion, we calculated the proportion of REM sleep time without atonia and correlated it to pallidal beta power. However, this correlation is not found to be significant (Spearman $\rho = 0.26$, $P = 0.47$), despite that the severity of RSWA is moderately correlated with RBDSQ (Spearman $\rho = 0.55$, $P = 0.09$). These additional analyses suggest that the relation between pallidal beta power during NREM sleep and RBD could indeed be inconclusive and be confounded by general sleep disturbance. Therefore, and in reply

to reviewer 2, we moved these results to the supplementary information and discussed this point critically in the Discussion section.

Supplementary Fig. 5 Correlations between pallidal beta band power and REM sleep

behavior disorder severity ratings. The heatmap on the right shows that beta power in NREM (Spearman $\rho = 0.67$, $P = 0.02$), but not REM sleep is significantly correlated with the REM sleep behavior disorder-screening questionnaire (RBDSQ) score. However, no correlation is observed between beta power in NREM sleep and the time proportion/severity of REM sleep without atonia (RSWA) (Spearman $\rho = 0.26$, $P = 0.47$), despite that the severity of RSWA is moderately correlated with the RBDSQ score (Spearman $\rho = 0.55$, $P = 0.09$). Scatter plots for the abovementioned correlations are shown separately on the left.

In the Discussion section,

“In the present cohort, dream enactment was not observed in any of the included PD patients, but NREM but not REM related beta activity was found to be associated with higher scores in the RBD-Screening Questionnaire. Given that this questionnaire is unspecific and does not provide a continuous scaling of RBD severity, we further quantified the loss of REM sleep atonia, as an indirect marker of RBD severity. We observed REM sleep without atonia in a median of 26.1% of all REM sleep epochs in PD patients, but found no significant correlation of the relative proportion of these epochs with beta activity measures. Thus, considering the limited sensitivity and specificity of the RBD-Screening Questionnaire in

diagnosing REM sleep behavior disorder²⁹, we speculate that the correlation between NREM beta power and RBDSQ could rather corroborate our abovementioned finding that beta can generally index sleep disturbances, which is further substantiated by the high correlation between PSQI and RBDSQ scores ($\rho = 0.821, P = 0.001$).”

Since Reviewer #2 raises similar questions, we kindly refer Reviewer #3 to our response to Reviewer #2, where this point has been detailly elaborated.

5. Minor:

Findings and conclusions should not be mentioned in the introduction. I would suggest to end the introduction with the aims. Also, methods are reported in the first part of the results section and should be moved to the methods section.

Reply:

We have now revised the text accordingly and the end of the Introduction now reads:

“While modulation of beta activity was reported during sleep cycles, the pathophysiological impact of this activity pattern on sleep quality in human patients remains unknown. Recently, a potential relationship between beta activity and sleep disturbance has first been reported in a non-human primate model of Parkinson’s disease¹⁶, urging for further investigation in the human domain. With the present study, we aim to address this important knowledge gap by comparing pathological beta activity across sleep cycles in the internal pallidum from patients with Parkinson’s disease with a control group of subjects suffering from dystonia, a different neurological disorder treated with internal pallidum DBS.”

For the results section, descriptions of the functional role of the GPi are removed. Reporting on the medication ON/OFF state of included subjects is moved to the methods section. Now the first part of the results reads:

“Twelve subjects with Parkinson’s disease and twenty subjects with dystonia

undergoing DBS electrode implantation in the ventro-posterior-lateral (motor) domain of the internal pallidum (GPi) were recruited for electrophysiological recordings during sleep. Standardized polysomnography was combined with invasive local field potential (LFP) recordings from the DBS electrodes in the GPi (Fig. 1a). After visual and algorithmic sleep staging, data from 40 nights across 32 patients were analyzed (for clinical information see Table 1). A comparison of sleep parameters indicated that patients with PD had significantly shorter total sleep time, longer sleep latency, less REM sleep, and more sleep segmentations than patients with dystonia (Table 2). Representative non-REM (NREM) and REM sleep epochs as well as a whole-night hypnogram and the corresponding spectrogram (from subject PD-8) are shown in Fig. 1b-d. Average power spectra corroborate previous reports indicating reduced beta activity during non-REM¹¹⁻¹⁴, when compared to awake and REM stages (Fig. 1e-f). However, these studies did not compare beta activity during sleep to a control group without Parkinson's disease.”

6. I would suggest a language review for the discussion.

Reply:

A careful language review has been made by the senior authors throughout the discussion section.

Below are a few examples,

(1) Text in the original manuscript,

“Our findings corroborate an impactful report linking beta activity with sleep disturbance in the 1-methyl-4-phenyl-1,2,3,6-tetrahydropyridine (MPTP) non-human primate model of PD¹⁶. The study demonstrated that MPTP intoxication induced increased beta activity during NREM sleep across all basal ganglia hubs. Beta activity after MPTP resulted in elevated NREM beta power and spiking in the subthalamic nucleus, external and internal pallidum.”

And after revision,

“Our findings corroborate an impactful report linking beta activity with sleep disturbance in the 1-methyl-4-phenyl-1,2,3,6-tetrahydropyridine (MPTP) non-human primate model of PD¹⁶. The study demonstrated that MPTP intoxication induced elevated beta power and spiking during NREM sleep in the subthalamic nucleus and external and internal pallidum.”

(2) Text in the original manuscript,

“In human studies, such direct comparisons to healthy states are obviated by the fact that invasive recordings are only available in patients treated with DBS for brain disorders. To account for the relative changes in activity, control groups with different disorders targeted in the same nucleus are necessary²⁰⁻²².”

And after revision,

“In human studies, such direct comparisons to healthy states are not possible, because invasive recordings are only available in patients undergoing neurosurgical interventions for brain disorders. To relate neural activity to disease specific aspects, two strategies have emerged: a) relative differences in activity from the same anatomical structure can be compared to control groups with other brain disorders²⁰⁻²², and b) within cohort correlations may indicate associations of brain activity patterns with clinical signs of the disease²³.”

(3) Text in the original manuscript,

“Furthermore, the best stimulation pattern required to suppress beta during NREM remains to be determined. In addition, since we did not identify behavior correlates of REM sleep beta here, and given evidence showing that DBS may occasionally induce *de novo* RBD³⁵, it requires further investigation whether DBS stimulation should be opened or closed during REM sleep.”

And after revision,

“Furthermore, the best stimulation pattern required to suppress beta during sleep remains to be determined. Since we did not identify sleep-related behavior correlates of beta activity during REM stage here, and given evidence showing that

DBS may occasionally induce *de novo* RBD³⁶, it requires further investigation on whether stimulation should be switched on or off during REM sleep.”

7. In the last sentence of the limitation, I would suggest using "basal ganglia during sleep" instead of "sleeping basal ganglia", to avoid misinterpretations.

Reply:

This has been revised, we thank the reviewer’s constructive comments.

“A fourth limitation of the present study is the focus on excessive beta synchronization in PD. Future investigations may lay more focus on increased pallidal theta power during REM sleep in dystonia to expand our understanding of disease-specific oscillatory abnormalities in the basal ganglia during sleep.”

References

1. Mayer, G. *et al.* Quantification of tonic and phasic muscle activity in REM sleep behavior disorder. *J Clin Neurophysiol* **25**, 48–55 (2008).
2. Hertenstein, E. *et al.* Sleep in patients with primary dystonia: A systematic review on the state of research and perspectives. *Sleep Med Rev* **26**, 95–107 (2016).
3. Thompson, J. A. *et al.* Sleep patterns in Parkinson's disease: direct recordings from the subthalamic nucleus. *J Neurol Neurosurg Psychiatry* **89**, 95–104 (2018).
4. Collop, N. A. Scoring variability between polysomnography technologists in different sleep laboratories. *Sleep Med* **3**, 43–47 (2002).

REVIEWERS' COMMENTS

Reviewer #1 (Remarks to the Author):

I consider that the authors have responded to all my comments. I have no more questions.

Reviewer #2 (Remarks to the Author):

The authors have satisfactorily addressed all issues raised

Reviewer #3 (Remarks to the Author):

Comments have been addressed, and the manuscript improved after revisions.

I have a single further comment: the authors should specify if RWA was scored according to international guidelines, as they refer to the IRBDSG guidelines only when stating they scored 3s mini-epochs, but refer to ref. 48 for what concerns RWA scoring. Please specify.

Reviewer #1 (Remarks to the Author):

I consider that the authors have responded to all my comments. I have no more questions.

Reply:

We thank the reviewer for the assistance in improving the manuscript.

Reviewer #2 (Remarks to the Author):

The authors have satisfactorily addressed all issues raised

Reply:

We thank the reviewer for the assistance in improving the manuscript.

Reviewer #3 (Remarks to the Author):

Comments have been addressed, and the manuscript improved after revisions.

I have a single further comment: the authors should specify if RWA was scored according to international guidelines, as they refer to the IRBDSG guidelines only when stating they scored 3s mini-epochs, but refer to ref. 48 for what concerns RWA scoring. Please specify.

Reply:

We thank the reviewer's suggestion. The RWA was scored based on a previously established automated RWA quantification approach. We choose the automated, rather than manual, RWA quantification approach because automated/algorithm-based approaches alleviate potential bias introduced by human scorers and improve repeatability.

However, it should be admitted that the IRBDSG guidelines stated that "Currently it is not possible to recommend any single method for automatic scoring of RWA."¹ Instead, they listed several automated RWA quantification approaches that have previously been established. Some of them are included in commercial products only or not freely available though (i.e., the code cannot be downloaded). The one we chose was derived from a paper published in *Sleep* in 2007². The methods are well documented and can be readily replicated, and were proven to produce valid conclusions in their study. Therefore, we chose this approach.

We have now specified this point in the Methods section:

"RSWA was determined when the chin EMG (10-70 Hz bandpass) variance in REM sleep was higher than two times the 5th percentile of the chin EMG variance in NREM sleep, which is an established approach listed in the International RBD Study Group (IRBDSG) guidance^{48,49}. For a better temporal resolution and as recommended by the IRBDSG⁴⁹, the above determination was made for each 3-second REM sleep segment."

References

1. Cesari, M. *et al.* Video-polysomnography procedures for diagnosis of rapid eye movement sleep behavior disorder (RBD) and the identification of its prodromal stages: guidelines from the International RBD Study Group. *Sleep* **45**, zsab257 (2022).
2. Burns, J. W. *et al.* EMG variance during polysomnography as an assessment for REM sleep behavior disorder. *Sleep* **30**, 1771–1778 (2007).